# Development of explicit definitions of potentially inappropriate prescriptions for antidiabetic drugs in patients with type 2 diabetes: A multidisciplinary qualitative study

Erwin Gerard[1]*, Paul Quindroit[1], Matthieu Calafiore[1,2], Jan Baran[2], Sophie Gautier[3], Stéphanie Genay[4,5], Bertrand Decaudin[4,5], Madleen Lemaitre[1,6], Anne Vambergue[6,7], Jean-Baptiste Beuscart[1]

1 CHU Lille, ULR 2694 - METRICS: Evaluation des Technologies de Santé et des Pratiques Médicales, Univ. Lille, Lille, France, 2 Department of General Practice, University of Lille, Lille, Lille, France, 3 CHU de Lille, UMR-S1172, Center for Pharmacovigilance, Univ. Lille, Lille, France, 4 CHU Lille, Institut de Pharmacie, Lille, France, 5 CHU Lille, ULR 7365 - GRITA - Groupe de Recherche sur les formes Injectables et les Technologies Associées, Univ. Lille, Lille, France, 6 Department of Diabetology, CHU Lille, Endocrinology, Metabolism and Nutrition, Lille University Hospital, Lille, France, 7 European Genomic Institute for Diabetes, University School of Medicine, Lille, France

* erwin.gerard@univ-lille.fr

**Data Availability Statement:** All relevant data are within the manuscript and its Supporting information files.

## Abstract

### Purpose

The management of type 2 diabetes mellitus patients has changed over the past decade, and a large number of antidiabetic drug treatment options are now available. This complexity poses challenges for healthcare professionals and may result in potentially inappropriate prescriptions of antidiabetic drugs in patients with type 2 diabetes mellitus which can be limited using screening tools. The effectiveness of explicit tools such as lists of potentially inappropriate prescriptions has been widely demonstrated. The aim was to set up nominal groups of healthcare professionals from several disciplines and develop a list of explicit definition of potentially inappropriate prescriptions of antidiabetic drugs.

### Methods

In a qualitative, nominal-groups approach, 30 diabetologists, general practitioners, and pharmacists in France developed explicit definitions of potentially inappropriate prescriptions of antidiabetic drugs in patients with type 2 diabetes mellitus. A nominal group technique is a structured method that encourages all the participants to contribute and makes it easier to reach an agreement quickly. Each meeting lasted for two hours.

### Results

The three nominal groups comprised 14 pharmacists, 10 diabetologists, and 6 general practitioners and generated 89 explicit definitions. These definitions were subsequently merged and validated by the steering committee and nominal group participants, resulting in 38

**Funding:** This research was funded by PreciDIAB, which is jointly supported by the French National Agency for Research (ANR- 18- IBHU- 0001), by the European Union (FEDER - agreement NP0025517), by the Hauts- de- France Regional Council (agreement 20001891/NP0025517) and by the European Metropolis of Lille (MEL, agreement 2019_ESR_11). The funder had no role in study design, data collection and analysis, decision to publish, or preparation of the manuscript.

**Competing interests:** The authors have declared that no competing interests exist.

validated explicit definitions of potentially inappropriate prescriptions of antidiabetic drugs. The definitions encompassed four contexts: (i) the temporary discontinuation of a medication during acute illness (n = 9; 24%), (ii) dose level adjustments (n = 23; 60%), (iii) inappropriate treatment initiation (n = 3; 8%), and (iv) the need for further monitoring in the management of type 2 diabetes mellitus (n = 3; 8%).

## Conclusion

The results of our qualitative study show that it is possible to develop a specific list of explicit definitions of potentially inappropriate prescriptions of antidiabetic drugs in patients with type 2 diabetes mellitus by gathering the opinions of healthcare professionals caring for these patients. This list of 38 explicit definitions necessitates additional confirmation by expert consensus before use in clinical practice.

## 1. Introduction

Over the last decade, the management of patients with type 2 diabetes mellitus (T2DM) has undergone many changes; for example, a number of new antidiabetic drugs (ADs, including SGLT-2 inhibitors, GLP-1 receptor agonists, and DPP-4 inhibitors) have been approved as first-line treatments (either as monotherapies or in combination with other drugs) [1–5]. Managing diabetes is a challenge for all healthcare professionals, including those who are not diabetes specialists or when patients are managed in an isolated, non-multidisciplinary context. Therefore, the risk of potentially inappropriate prescriptions (PIPs) in patients with T2DM is likely to have risen. Improving the appropriateness of prescriptions in routine practice can also be challenging.

A professional consensus and clinical practice guidelines on the follow-up of patients with diabetes have been updated recently [6]. Patients with T2DM are mainly followed up by general practitioners [7–11]. However, various studies have shown that even after these recent updates, compliance with the guidelines is suboptimal–notably for patients followed up only by general practitioners [12–14].

The assessment of the appropriateness of AD prescriptions is generally based on an implicit approach, referencing to guidelines or medical practice [15, 16]. This approach is defined as implicit because it requires an expert's assessment of the quality of care in relation to the patient's condition and the medical literature [17, 18].

The Medication Appropriateness Index is currently the most widely used implicit method [18, 19]. However, implicit approaches have practical limitations because their application is time-consuming and requires specialized knowledge and skills.

It is also possible to reduced PIPs by applying an explicit approach, i.e. predefined explicit criteria that combine the various items of information related to the prescription in the absence of an expert assessment (e.g. a prescription of metformin in a patient with an estimated glomerular filtration rate (eGFR) below 30 mL/min/1.73 m$^2$ is a PIP) [19, 20]. The value of explicit approaches for the detection PIPs has been well documented [21–24].

Some experts recommend that implicit and explicit approaches should be combined for optimal patient management [24, 25]. With the increasing use of computerized physician order entry and electronic medical records, the explicit approach becomes more practical and easier to implement compared to integrating implicit criteria [24, 26–30]. Explicit criteria can

be coded and integrated into a clinical decision support system, in order to systematically flag up PIPs to both non-specialist and specialist prescribers [20, 25, 31–34].

A recent systematic review of the literature recorded a total of 56 explicit definitions of PIPs of ADs (hereafter referred to as AD-PIPs) but highlighted a lack of consensus; the definitions were heterogenous and focused primarily on the at-risk situations related to (i) biguanide prescriptions in patients with renal dysfunction and (ii) the prescription of sulfonylureas to older adults [35]. Furthermore, the reviewed definitions were essentially derived from other explicit tools that are not specific to T2DM (such as the STOPP/START and Beers criteria [36, 37]) or from research that did not systematically involve clinicians. However, the routine management of patients with T2DM involves healthcare professionals from different disciplines and medical specialties (such as general practitioners, diabetologists, and pharmacists), and we reasoned that their opinions on PIP-ADs would be highly relevant [38, 39]. The nominal group technique has been widely used to gather the opinions of healthcare professionals and to generate new ideas [40–45].

Hence, the objective of the present study was to develop a list of explicit definition of PIP-ADs, based on the opinions of healthcare professionals from different disciplines.

## 2. Materials and methods

### 2.1. Study design

Our method for developing explicit definitions of PIP-ADs in patients with T2DM has been described previously [46]. The present work involved a qualitative approach that had already been used to set up explicit definitions of PIP-ADs in hospitalized older patients [44]. Here, we applied a qualitative, nominal-group-based technique [40–43] with expert diabetologists, general practitioners and pharmacists in France. A steering committee (comprising a diabetologist, a general practitioner, a clinical pharmacist, a community pharmacist, and a pharmacologist) was set up to validate the methodology and to monitor the study's progress. The present report complied with the Consolidated Criteria for Reporting Qualitative Research (COREQ) [47] (S1 Table). These checklists relate to the sampling method, data collection setting, data collection method, respondent validation of findings, method of recording data, description of the derivation of themes, and inclusion of supporting quotations. The items are grouped into three domains: (i) research team and reflexivity, (ii) study design and (iii) data analysis and reporting.

### 2.2. Study objective

The objective of the present study was to develop a list of explicit definitions of PIP-ADs in patients with T2DM; we reasoned that such a list might help physicians to prescribe ADs more effectively and safely.

**2.2.1. Explicit definitions.** As described in the Introduction, two different types of approach can be used to evaluate the (in)appropriateness of drug prescriptions in clinical pharmacology: (i) so-called "implicit" approaches based on expert judgements of the quality of care with regard to the patient's situation and guidelines on the use of drugs, and (ii) explicit approaches based on predefined criteria for the analysis of drug prescriptions and that do not require intervention by an expert.

**2.2.2. Potentially inappropriate prescription.** Explicit definitions cover situations considered by experts to be generally inappropriate, as defined in the literature or by expert consensus. However, when an explicit definition is applied to a given prescription, the absence of expert opinion means that the prescription's inappropriateness cannot be confirmed. Therefore, explicit definitions correspond to PIPs.

## 2.3. Recruitment of participants

We set up three nominal groups with 5 to 15 participants each following the recommendations of Allen et al. [41] and McMillian et al. [40]. The participants were recruited by e-mail, with help from the steering committee members.

Recruitment was started on 10[th] March 2022. Recruitment was completed the day before the corresponding nominal group. If there were not enough participants, a nominal group could be rescheduled for another date.

The first group was composed of pharmacists (hospital pharmacists and community pharmacists), the second was composed of diabetologists, and the third was composed of general practitioners. The study investigators had no contact with the nominal groups members prior to the study. The participants' characteristics (age, sex, year of qualification, specialty, and involvement in training on AD stewardship) were recorded.

This study was approved by the University of Lille Research Ethics Committee. Participants were informed of the study purpose, questions, and procedure prior to the nominal groups. Verbal consent was obtained at the beginning of both nominal groups because the study presented no more than minimal risk of harm to subjects and involved no procedures for which written consent is normally required outside of the research context.

## 2.4. The nominal groups' workflow

The nominal groups met in Lille (France) three times: on May 5[th] and May 10[th], 2022, and on January 11[th], 2023. Each meeting lasted for two hours. Two investigators were present (EG, a PharmD studying for a PhD, and PQ, a RN and PhD scientist) and acted as a facilitator and an observer, respectively. At the start of the meeting, participants viewed a presentation of study's background and objectives. Each participant then considered the questions individually and suggested definitions of PIP-ADs. This process continued until the participants had run out of suggestions. The suggested definitions were recorded on a computer spreadsheet (Excel®, Microsoft Corporation, Redmond, WA, USA) and then shown to the participants. Next, each definition was discussed, and the latter's explicit nature was confirmed by the group. If necessary, a definition could be reformulated. New definitions could also be added at this stage.

## 2.5. The merger of definitions suggested by the nominal groups

In order to eliminate duplicates within and between groups, the suggested definitions were reviewed by two investigators (EG and PQ). After the removal of duplicates and the merger of similar definitions (e.g. the terms "nausea", "vomiting" and "diarrhoea" have been merged into "acute digestive disorders"), a single list of definitions was obtained. Any disagreements were discussed and resolved by consensus with a third researcher (JBB).

## 2.6. Validation of the list of explicit definitions

The list of explicit definitions was validated by the members of the steering committee and the members of the three groups. The included definitions had to be (i) explicit, (ii) related to ADs (excluding insulin) in the Anatomical Therapeutic Chemical class A10B and in the context of T2DM. If necessary, a definition could be reformulated. We excluded definitions that (i) were implicit, (ii) were outside the scope of the study, (iii) had already been stated in the American Diabetes Association and Société francophone du diabète guidelines and, (iv) were less relevant with regard to the summary of product characteristics "dosing regimens" or "special warnings and precautions for use" (e.g. minimum and maximum dose levels, as well as the times at which they should be taken). The members of the steering committee discussed each

suggested definition and drew up a list of validated definitions. Lastly, the list of explicit definitions was submitted to all the nominal group members for final validation.

## 2.7. Classification of definition

PIPs are typically presented in the following format: "it is (always) inappropriate to prescribe a drug X in a situation Y (e.g. a patient aged 75 or over)" [36, 37]. However, the members of the nominal groups suggested definitions that appeared to be non-fixed or that depended on the context. Four contexts were identified by three investigators (EG, PQ, and JBB) and validated by the steering committee: (i) the potential need to temporarily discontinue a medication in the event of an acute illness; (ii) the potential need to adjust a dosage regimen in a chronic disease context for example; (iii) inappropriate treatment initiation; and (iv) the need for additional monitoring in the management of T2DM.

## 3. Results

### 3.1. Nominal groups and the characteristics of the group members

The three nominal groups comprised 30 members in total: 14 in the nominal group of pharmacists, 10 in the nominal group of diabetologists, and 6 in the nominal group of general practitioners (Table 1).

### 3.2. Numbers of explicit definitions suggested by the nominal groups

The nominal group of pharmacists generated 179 suggestions, which were merged into 71 definitions (Fig 1). The nominal group of diabetologists generated 119 suggestions, which were merged into 43 definitions. Lastly, the nominal group of general practitioners generated 70 suggestions, which were merged into 28 definitions. The three lists were then merged into a single list of 89 definitions of PIP-ADs in patients with T2DM.

### 3.3. Validation of explicit definitions

Six of the 89 definitions (6.7%) were deemed to be non-explicit; for example, the definition mentioned anorexia without specifying the type and cause of anorexia or giving an explicit definition of anorexia. Eleven definitions (12.4%) were deemed to be outside the scope of the study; for example, one definition was linked to intensive care and a metformin overdose, which is an adverse drug event and not a PIP. Eighteen (20.2%) were deemed to be less relevant because they were related to items already described in the summary of product characteristics "dosing regimens" or "special warnings and precautions for use" sections (e.g. the minimum and maximum dose levels, and the times at which the AD should be taken). Explicit definitions related to items in the "contraindication" section were not excluded.

During the study period, the American Diabetes Association and Société francophone du diabète published explicit rules on the treatment of diabetes in patients with chronic kidney

Table 1. Characteristics of the nominal group members.

| Characteristics | Participants n = 30 (100.0%) | General practitioners n = 6 (20.0%) | Diabetologists n = 10 (33.3%) | Pharmacists n = 14 (46.6%) |
|---|---|---|---|---|
| Age (median (range)) | 42.5 (27; 67) | 48.5 (40; 67) | 47.0 (29; 59) | 39.5 (27; 59) |
| Years since qualification (MD or PharmD, median (range)) | 15 (2; 41) | 22 (9; 40) | 20 (1; 31) | 9 (1; 37) |
| Females | 17 (56.6%) | 0 (0.0%) | 6 (60.0%) | 11 (78.6%) |
| Involved in antidiabetic drug stewardship | 15 (50.0%) | 5 (83.3%) | 8 (80.0%) | 2 (14.3%) |

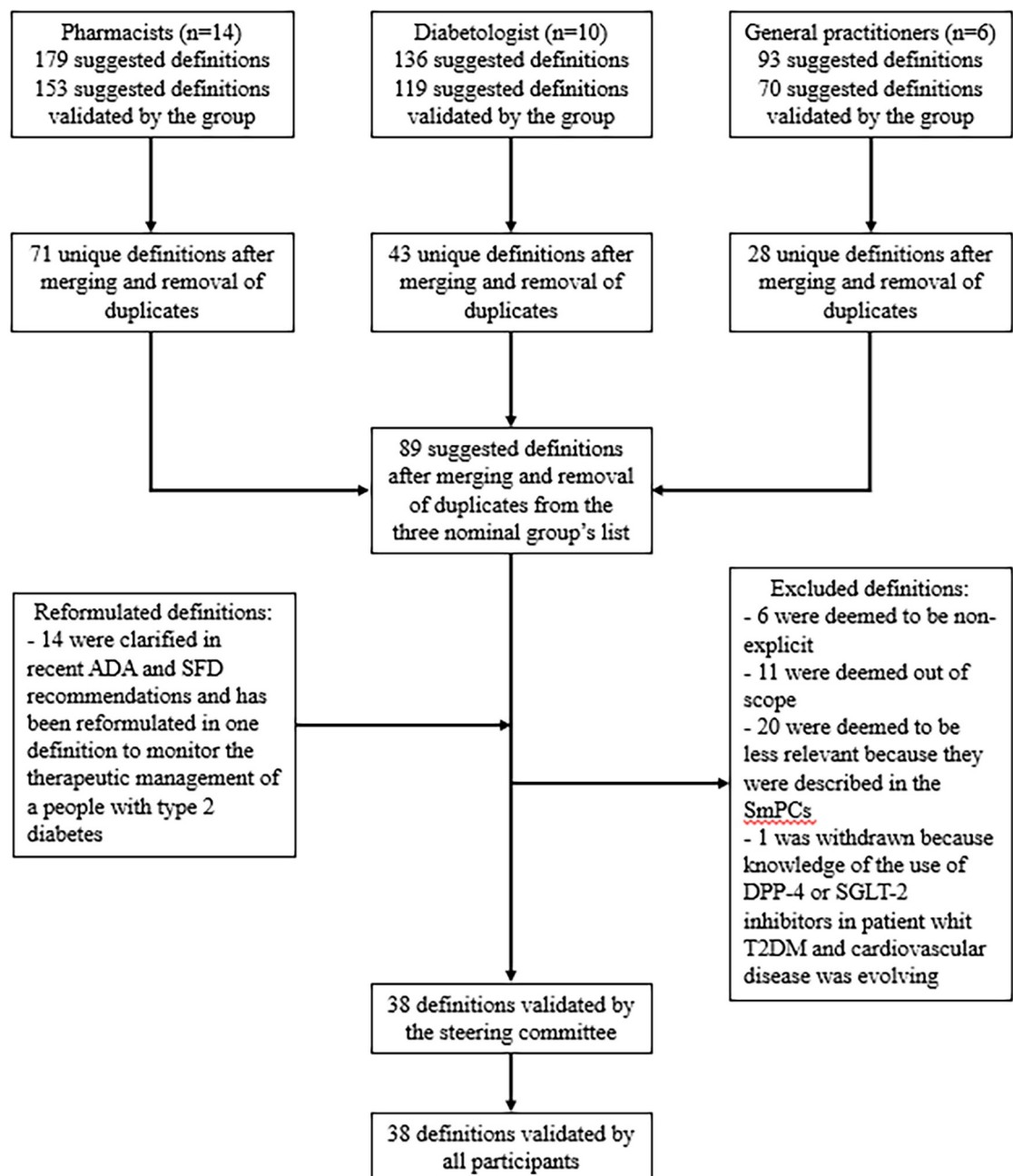

**Fig 1. Study flow chart.** ADA = American diabetes association; DPP-4 = dipeptidyl peptidase-4; SFD = Société francophone du diabète; SGLT-2; sodium-glucose cotransporter-2; T2DM = Type 2 diabetes mellitus.

disease [48, 49]. For example, the Société francophone du diabète guideline explicitly states that metformin can be prescribed to patients with an eGFR between 59 and 30 ml/min/m$^2$ but is not recommended for patients with an eGFR below 30 ml/min/m$^2$. As a result, 14 (15.7%) definitions described the adjustment of treatment with regard to the eGFR.

We identified and classified the validated explicit definitions in the following contexts: (i) the potential need to temporarily discontinue a medication in the event of acute illness (n = 9;

23.6%), (ii) the potential need to adjust the dosage regimen (n = 23; 60.5%), (iii) inappropriate treatment initiation (n = 3; 7.9%), and (iv) the need for additional monitoring in the management of a patient with T2DM (n = 3; 7.9%) (Table 2).

### 3.4. The final list of PIP-ADs

A final list of 38 explicit definitions was submitted to the nominal group members for validation (Table 2). All the definitions and their classification were validated by the steering committee and the nominal group members. Six Anatomical Therapeutic Chemical classes were mentioned at least once: biguanides, SGLT-2, inhibitors, GLP-1 receptor agonists, DPP-4 inhibitors, sulfonylureas, and repaglinide.

Most of definitions covered all six classes of AD (n = 22; 6%), including one specific definition for injectable ADs ("It may be necessary to reassess and adjust the prescription of an injectable AD if it is difficult to use this administration route (e.g. in a patient with a fear of injections)") and one specific definition for oral ADs ("It may be necessary to reassess and adjust the prescription of oral ADs in a patient who has difficulty swallowing pills"). Most of the definitions related to a single drug class concerned sulfonylureas and repaglinide (n = 7; 18.4%). Other definitions were related to GLP-1 receptor agonists (n = 4, 10.5%), SGLT-2 inhibitors (n = 4, 10.5%), biguanides (n = 2; 5.2%), and DPP-4 inhibitors (n = 2; 5.2%).

## 4. Discussion

### 4.1. Main findings

The present study listed 38 explicit definitions of PIP-ADs in patients with T2DM. To the best of our knowledge, this qualitative study of clinicians is the first to have generated a list of explicit definitions of PIPs specifically in patients with T2DM. The nominal group method enabled 30 healthcare professionals from various disciplines and specialties to generate and classify definitions. However, the selected definitions must be validated by expert consensus (e.g. in a Delphi survey) before they can be used in routine clinical practice [40, 43, 50].

Our previous systematic review highlighted the variable levels of precision and written formulation of explicit definitions [35]. Our present work focused on the management of patient with T2DM, whereas most published lists of explicit criteria were related to older patients [51]. Indeed, Succurro et al. highlighted poor adherence to current guidelines on diabetes management [52], and a meta-analysis by Mahmoud et al. identified various factors to be considered when deciding on the most appropriate ADs for patients with T2DM [53]. Screening tools based on explicit criteria have proven their worth in reducing PIPs [54]. In particular, these tools could help non-specialists, whose lack of training [55] or reluctance to alter prescriptions by colleagues can constitute barriers to appropriate prescribing [56].

### 4.2. The need for validation by expert consensus

Explicit definitions of PIP-ADs listed by nominal groups have not been approved by an expert consensus and must be validated in a Delphi survey [57]. The objective of the present qualitative study was to draw up a rigorous list of explicit definitions. However, the study's methodology prevented us from modifying the recorded content; for example, some of the content was suggested by a single participant might not considered to be relevant by all clinicians. The list of explicit definitions of AD-PIPs obtained in a literature review and the list generated by the present qualitative study should be merged. The merged list should then be submitted to a panel of experts, assessed for relevance, and then agreed by consensus.

**Table 2. List of validated explicit definitions of AD-PIPs (excluding insulin) in patients with T2DM.** It must be borne in mind that these definitions require external validation in a Delphi survey before they can be used in practice.

| |
|---|
| **It may be necessary to temporarily discontinue**. . . |
| the prescription of ADs in a patient requiring general anaesthesia for surgery. |
| the prescription of ADs in a patient with an acute digestive disorder (nausea, diarrhoea, vomiting, etc.) |
| the prescription of ADs in a patient with acute liver injury |
| the prescription of ADs in a patient with signs and/or symptoms of infection (such as fever, sepsis, the prescription of antibiotics, etc.). |
| the prescription of ADs in a patient with a diagnosis of acute pancreatitis, including suspected acute pancreatitis (elevated lipase levels, abdominal pain, etc.). |
| the prescription of an SGLT-2 inhibitor in a patient with a skin infection (an abscess, Verneuil's disease, etc.) |
| the prescription of an SGLT-2 inhibitor in a patient with unstable grade 4 peripheral arterial disease |
| the prescription of metformin for 48 hours when a patient requires an injection of iodinated contrast medium |
| the prescription of repaglinide or sulfonylureas in a patient with a capillary blood glucose level < 0.8g/L at the time of taking the medication |
| **It may be necessary to not initiate**. . . |
| ADs in a patient whose diagnosis of type 2 diabetes is uncertain (2 venous glycaemia measurements of at least 1.26 g/L or 1 venous glycaemia measurement of at least 2 g/L) |
| Repaglinide or sulfonylureas in a patient aged over 75 |
| Repaglinide or sulfonylureas in a patient with an Hb1Ac level < 7% |
| **It may be necessary to reassess and adjust**. . . |
| the prescription of ADs in a patient with a chronic digestive condition (Crohn's disease, ulcerative colitis, etc.) |
| the prescription of ADs in a patient aged over 75 |
| the prescription of ADs in a patient who does not tolerate the treatment well |
| the prescription of ADs in a dependent patient (e.g. with a movement disorders or cognitive disorder) who does not receive from home care or who is not in an institution |
| the therapeutic strategy in two- or three-drug combination treatments involving repaglinide or sulfonylureas in a patient with an HbA1c level < 7%. |
| the prescription of ADs in a patient with a history of ketoacidosis |
| the prescription of ADs in a patient with a chronic liver condition |
| the prescription of ADs in a patient with a chronic pancreatic condition or a history of acute pancreatitis |
| the prescription of ADs if refusal of this treatment is clearly stated in the patient's medical records |
| the prescription of ADs for a patient taking more than eight prescription drugs daily and with an HbA1c level < 7% |
| the prescription of a combination of two ADs from the same drug class |
| the prescription of injectable ADs if it is difficult to use this administration route (e.g. in a patient with a fear of injections) |
| the prescription of oral ADs in a patient who has difficulty swallowing pills |
| the prescription of GLP-1 receptor agonist in a patient with a history of gallstones |
| the prescription of metformin, a GLP-1 receptor agonist or a DPP4 inhibitor in a patient with a 1.5-fold increase in creatinine in 7 days |
| the combination of a GLP-1 receptor agonist and a DPP4 inhibitor |
| the prescription of an SGLT-2 inhibitor in a patient with past or ongoing genito-urinary tract involvement (a lower urinary tract infection in men or recurrent low urinary tract infections (4 or more times over 12 months) in women), upper urinary tract infection, micturition disorder, genital mycosis, etc.) |
| the prescription of an SGLT-2 inhibitor or a GLP-1 receptor agonist in a patient with a body mass index < 18 kg/m$^2$ |
| the prescription of metformin in a patient with a condition that may lead to hypoxia (respiratory pathology, sleep apnoea syndrome, oxygen therapy, etc.) |
| the combination of repaglinide and sulfonylureas |
| the prescription of repaglinide or sulfonylureas in a patient with a hypocaloric diet (with weight loss) |
| the prescription of repaglinide or sulfonylureas in a patient reporting symptomatic hypoglycaemia (malaise, sweating, palpitation, intense hunger, dizziness, "jelly legs" etc.). |
| the prescription of repaglinide or sulfonylureas in a patient with a body mass index > 30 kg/m$^2$. |

(*Continued*)

**Table 2.** (Continued)

| **It may be necessary to monitor**... |
| --- |
| the eGFR in a patient with T2DM at least once a year |
| the HbA1c level in a patient with T2DM at least once a year |
| the patient's treatment adherence and involvement in a patient education programme |

## 4.3. Perspectives for applying explicit definitions of PIP-ADs

A recent literature review emphasized the value of using explicit tools to detect PIPs [58]. Moreover, explicit definitions could advantageously be included in clinical decision support system, with a view to the automated detection of at-risk situations in routine practice [59, 60] and an increase in the proportion of PIPs detected [33]. A recent review highlighted the role of clinical decision support system in improving high quality medication use and enhancing the quality of care [61]. Interventions within the scope of pharmacists' practice span a variety of domains, including dynamic monitoring, ensuring drug safety for patients with renal impairment, addressing medication safety concerns related to QT prolongation, performing dosing calculations, conducting medication reconciliations, and overseeing general medication usage and safety. However, the effectiveness of a clinical decision support system based on PIP definitions would have to be assessed in the context of care (primary care, hospital care, etc.) [62]. Some of the explicit definitions of PIP-ADs mentioned in our study may be more relevant in a hospital setting, whereas others may be more applicable to primary care. It will also be necessary to target the practitioners for whom these definitions will be useful: some definitions are certainly intended for general practitioners, with others for pharmacists, and yet others for nurses [63]. Two approaches could be used to assess the relevance of these definitions, once validated in a Delphi survey. Firstly, the definitions could be used in clinical decision support system, and their relevance to clinical practice could be evaluated [59]. Secondly, one could reuse data from a health data warehouse and apply data-mining methods [64, 65].

## 4.4. Strengths and limitations

Our study methodology made it possible to gather suggestions from 30 healthcare professionals who are direct in contact with patients and prescriptions, i.e. general practitioners, pharmacists and diabetologists. Each part of the merging, validation and classification processes was carried out independently by two investigators, and any disagreements were discussed and resolved by consensus with a third researcher. All steps were discussed and validated by a steering committee. At the end of the study, the final list of definition was validated by all the nominal group members. We completed 26 of the 32 COREQ items: six items were not applied because they were not applicable to the nominal group technique. However, the COREQ checklist was drawn up to help researchers to report important aspects of qualitative studies in general and focus groups and interviews in particular; in contrast, our qualitative study was based on the nominal groups technique [42].

Our study also had some limitations. All participants are based in France, the scope of some of the definitions proposed here may be limited. In particular, these definitions do not include diabetes therapies that are not prescribed in France, such as thiazolidinediones. Our study was qualitative; thus, our reporting did not depend on the number of times a given definition was mentioned in the nominal groups or by other study participants. The definitions reported here must be validated by expert consensus (e.g., in a Delphi survey) and cannot be used in practice

at this stage. Lastly, the applicability of explicit definitions and the potential for implementation in computer systems will need to be addressed in further research.

## 5. Conclusion

Our qualitative study involved 30 healthcare professionals (10 diabetologists, 6 general practitioners, and 14 pharmacists) and generated 38 explicit definitions of PIP-ADs classified with regard to four contexts: (i) the potential need to temporarily discontinue a medication in the event of acute illness, (ii) the potential need to adjust the dosage regimen, (iii) inappropriate treatment initiation, and (iv) the need for additional monitoring of a patient with T2DM. This list of 38 explicit definitions necessitates additional confirmation by expert consensus before use in clinical practice.

## Supporting information

**S1 Table. COREQ checklist.**
(PDF)

## Acknowledgments

We thank all the participants in the nominal groups. We thank David Fraser (Biotech Communication SARL, Ploudalmézeau, France) for copy-editing assistance.

## Author Contributions

**Conceptualization:** Erwin Gerard, Paul Quindroit, Jean-Baptiste Beuscart.

**Methodology:** Erwin Gerard, Paul Quindroit, Jean-Baptiste Beuscart.

**Project administration:** Jean-Baptiste Beuscart.

**Supervision:** Matthieu Calafiore, Jan Baran, Anne Vambergue, Jean-Baptiste Beuscart.

**Validation:** Matthieu Calafiore, Jan Baran, Sophie Gautier, Bertrand Decaudin, Madleen Lemaitre, Anne Vambergue, Jean-Baptiste Beuscart.

**Writing – original draft:** Erwin Gerard, Paul Quindroit, Jean-Baptiste Beuscart.

**Writing – review & editing:** Erwin Gerard, Paul Quindroit, Matthieu Calafiore, Jan Baran, Sophie Gautier, Stéphanie Genay, Bertrand Decaudin, Madleen Lemaitre, Anne Vambergue.

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
