## [Decision Letter · Decision Letter 0]

10 Apr 2024

PONE-D-24-07859Development of explicit definitions of potentially inappropriate prescriptions for antidiabetic drugs in patients with type 2 diabetes: a multidisciplinary, nominal-groups approach.PLOS ONE

Dear Dr. Gerard,

Thank you for submitting your manuscript to PLOS ONE. After careful consideration, we feel that it has merit but does not fully meet PLOS ONE’s publication criteria as it currently stands. Therefore, we invite you to submit a revised version of the manuscript that addresses the points raised during the review process.

We look forward to receiving your revised manuscript.

Kind regards,

Sairah Hafeez Kamran, PhD

Academic Editor

PLOS ONE

Journal Requirements:

2. Thank you for including your ethics statement:  "Participants were informed of the study purpose, questions, and procedure prior to the nominal groups. Verbal consent was obtained at the beginning of both nominal groups because the study presented no more than minimal risk of harm to subjects and involved no procedures for which written consent is normally required outside of the research context.".   

3. We noticed a mistake in your MS regarding recruitment dates. Please ensure this is corrected.

Reviewers' comments:

Reviewer's Responses to Questions

**Comments to the Author**

1. Is the manuscript technically sound, and do the data support the conclusions?

Reviewer #1: Partly

Reviewer #2: Yes

2. Has the statistical analysis been performed appropriately and rigorously? 

Reviewer #1: N/A

Reviewer #2: Yes

3. Have the authors made all data underlying the findings in their manuscript fully available?

Reviewer #1: Yes

Reviewer #2: Yes

4. Is the manuscript presented in an intelligible fashion and written in standard English?

Reviewer #1: Yes

Reviewer #2: Yes

5. Review Comments to the Author

Reviewer #1: March 14, 2029

PLOLS One review

To the authors:

This is an interesting paper on an important topic—preventing inappropriate drug administrations to persons with Type 2 DM. The paper has broad implications—and seeks to develop a T2DM standard that parallels the Beers criteria for geriatric pharmaceuticals. It includes results from a nominal group process in France consisting of pharmacists, general practitioners, and diabetologists who generated 89 explicit definitions. The claim is that this is the first to have produced a list of explicit definitions of potentially inappropriate prescriptions of T2DM drugs- and suggests that a Delphi process could validate the findings.

However, there are several important concerns:

Title- few readers will know what a nominal group approach is…I would reserve this to the abstract and to the text where this team can be defined.

Cover page- who funded the project? The cover indicates that there is no conflict of interest.

Abstract:

Purpose—has the management significantly changed or has there been an introduction of numerous new drugs and if so—could you provide an estimate of how large “significant changes” are with respect to number of new drugs for T2DM?

How does one know that screening tools could limit the numbers of potentially inappropriate drugs? Could the screening tools make matters more confusing—is there precedent for screening tools to improve rather than harm physician ordering (i.e by creating confusion and costing time)?

Methods: COREG should be in the paper, not necessarily in the abstract. Nominal group should be defined in the methods section of the abstract.

Results: is there any information on validity and reliability of the definitions. Did the steering committee vote on each definition?

Conclusion: this is very overstated- the conclusion should be that this is a great pilot effort- and should be redone in more settings in France for validation and potential pilot data. Is a Delphi panel the next step or is a program that included more nominal groups needed?

Abbreviations- there are too many here….Please try to decrease the number of abbreviations.

Manuscript

Introduction- again—too many abbreviations. The description of implicit and explicit needs more context and references. Many studies formally compare explicit and implicit evaluations and should be cited…These studies discuss pros and cons of each method.

Methods

The context for study design needs more descriptions as well--- in the introduction or methods. More text on when to use COREQ is also needed. Is there background about how many persons should be in the nominal groups and how many should be on the steering committee. Are there previous studies that have used the nominal group methods—what did they do and what did they find?

Objective

Why move directly to explicit—why not do implicit and explicit and then compare findings?

Recruitment—why 5 to 15 persons per group? How random was the selection process? Is the process of selection biased- and will this affect the results? How was the group makeup determined? Was there any training? Had the facilitators prior experience with nominal groups? Has the team succeeded in nominal group explicit checklist development?

What is an explicit vs implicit definition?

Results

It would be preferable to included four columns in the table—one for each of the three groups and then the overall characteristics (that are shown).

The table is clear.

Discussion

Findings—does the nominal literature state that a list that is generated needs to be externally validated? Why not compare explicit versus implicit now? This is a nice start, but several limitations are noted—some by the authors themselves?

I would have preferred to see possibly a second independent effort done and then compare the final results of each effort.

This reads more like a QI effort than a research study.

Reviewer #2: Overall, this manuscript is a well-written qualitative study. The choice of the study design was justified adequately. The methods, results, and discussion parts included a detailed procedure, well-designed tables, and a thorough interpretation of the findings which clearly answered the research questions.

Minor comments:

1. In the introduction, add more findings to discuss the significance of your study, especially why non-specialist physicians would prescribe AD prescriptions for T2DM patients and why it’s an issue in your country and globally?

2. In the introduction, discuss the most updated version of ADA guidelines to treat T2DM patients, and the limitations of the implicit approach to T2DM treatment.

3. The conclusion needs revision to highlight your important findings.

6. PLOS authors have the option to publish the peer review history of their article (what does this mean?). If published, this will include your full peer review and any attached files.

Reviewer #1: No

Reviewer #2: No

---

## [Author Response · Author response to Decision Letter 0]

29 May 2024

Response to Journal Requirements:

We thank Academic Editor for comments and constructive suggestions. We have carefully revised the manuscript according to each comment and suggestion.

Thank you for your comment. We have taken the necessary steps to correct the elements that were not correctly entered in accordance with PLOS ONE's style requirements.

Modifications: In the title page, we have removed the physical address of the corresponding author and specified his initials line 14 of the revised manuscript with track changes.

According to the PLOS ONE style templates,

• We have used a font size of 18pt font for all level 1 headings and 16pt fond for level 2 headings.

• We have changed the style of references by using square brackets instead of parentheses.

• We have deleted lines 17 to 19 of the revised manuscript with track changes specifying keywords and competing interests.

• On line 207 of the revised manuscript with track changes, we have rewritten fig 1 instead of figure 1.

• In the online submission system, we have changed the file "Figure_1.tiff" to "Fig1.tiff" and the file "Supplementary_information_S1" to S1 Table.pdf".

2. Thank you for including your ethics statement: "Participants were informed of the study purpose, questions, and procedure prior to the nominal groups. Verbal consent was obtained at the beginning of both nominal groups because the study presented no more than minimal risk of harm to subjects and involved no procedures for which written consent is normally required outside of the research context.". 

Please amend your current ethics statement to include the full name of the ethics comité/institutional review board(s) that approved your specific study. 

Modifications: We have amended our current ethics statement to include the full name of the ethics committee, and we have amended this statement in the Methods section of the manuscript, line 158 in the revised manuscript with track changes "This study was approved by the University of Lille Research Ethics Committee"; and we have added the same text to the "Ethics Statement" field of the submission form (via "Edit Submission").

3. We noticed a mistake in your MS regarding recruitment dates. Please ensure this is corrected.

Thank you for spotting this mistake. We have corrected it, line of the revised manuscript with track changes.

Modification: In line 150 of the revised manuscript with track changes, we have changed " 2022 " instead of " 2024 ".

We have corrected the indicated items. We apologize for any errors in translation.

This research was funded by PreciDIAB, which is jointly supported by the French National Agency for Research (ANR- 18- IBHU- 0001), by the European Union (FEDER - agreement NP0025517), by the Hauts- de- France Regional Council (agreement 20001891/NP0025517) and by the European Metropolis of Lille (MEL, agreement 2019_ESR_11).

The funder had no role in study design, data collection and analysis, decision to publish, or preparation of the manuscript. 

Response to Reviewer 1 Comments

We thank Reviewer 1 for comments and constructive suggestions. We have carefully revised the manuscript according to each comment and suggestion. In the following, our responses are given in blue.

Point 1. Title- few readers will know what a nominal group approach is…I would reserve this to the abstract and to the text where this team can be defined.

Thank you for your comment. We have changed this part of the title to read: "a multidisciplinary qualitative study".

Modification (title): Development of explicit definitions of potentially inappropriate prescriptions for antidiabetic drugs in patients with type 2 diabetes: a multidisciplinary qualitative study.

Point 2. Cover page- who funded the project? The cover indicates that there is no conflict of interest.

Project funding has been declared on the plosOne submission platform. The authors declare that they have no financial or non-financial conflicts of interest.

This project is part of the PreciDIAB project of the National Center for Precision Diabetic Medicine, which brings together several teacher-researchers and practitioners from the University of Lille and the Lille University Hospital. More information: https://www.precidiab.org/en/

The funder had no role in study design, data collection and analysis, decision to publish, or preparation of the manuscript.

Modifications: We deleted line 19 of the revised manuscript with track changes specifying competing interests according to the PLOS ONE style templates (see responses to Editor’s comments).

Point 3. Abstract - Purpose—has the management significantly changed or has there been an introduction of numerous new drugs and if so—could you provide an estimate of how large “significant changes” are with respect to number of new drugs for T2DM?

Thank you to the reviewer for this comment. The word "significant" was not correct. As shown by the work of Chong et al. (2023) [https://doi.org/10.1002/kjm2.12800] and Galindo et al. (2022) [https://doi.org/10.1136/ bmjmed-2022-000372], the management of patients living with type 2 diabetes has evolved over the last 10 years with new molecules, in particular SGLT-2 inhibitors, GLP-RAs and DPP4 inhibitors, in the choice of first-line treatments and combinations of therapies.

Modifications: In line 27 in the revised manuscript with track changes, we deleted the word “significant”. As a result, we have also removed the word “significant” from the introductory section, line 74.

Point 4. Abstract - How does one know that screening tools could limit the numbers of potentially inappropriate drugs? Could the screening tools make matters more confusing—is there precedent for screening tools to improve rather than harm physician ordering (i.e by creating confusion and costing time)?

Explicit criteria for potentially inappropriate prescribing (PIP) in older people have gained attention since the first publication of the Beers criteria in 1991. Explicit criteria are generally used as rigid standards and do not take into account individual differences among patients, nor the complexity and appropriateness of entire medication regimens [https://doi.org/10.1001/archinte.1991.00400090107019]. The effectiveness of explicit tools such as lists of potentially inappropriate prescriptions has been largely demonstrated. Several tools were used to assess PIM in hospitalized patients over 65most commonly Beer's criteria and the STOPP/START tool. In a systematic review of the literature, Alshammari et al. showed a reduction of PIM ranged from 3.5% up to 87% [https://doi.org/10.2147/DHPS.S303101].

We agree with the reviewer that for 15 years, screening tools implemented in clinical decision support systems (CDSS) have caused harm to physician ordering. This was due to several factors, including data quality and human factors [http://dx.doi.org/10.1016/j.jbi.2015.03.006]. However, CDSS have improved over the last decade and their effectiveness has been demonstrated in recent large RCTs [https://doi.org/10.1016/S0140-6736(23)02465-0 and https://doi.org/10.1001/jama.2024.6259].

Modifications: Lines 31-32 in the revised manuscript with track changes, the effectiveness of explicit tools such as lists of potentially inappropriate prescriptions has been widely demonstrated.

Point 5. Abstract - Methods: COREG should be in the paper, not necessarily in the abstract. Nominal group should be defined in the methods section of the abstract.

We agree with the reviewers. We have corrected the sentence.

Modifications: 

Lines 39-40 in the revised manuscript with track changes, the sentence mentioning COREQ criteria has been removed from the abstract. 

Lines 126-130 in the revised manuscript with track changes, the COREQ criteria has been defined in the study design paragraph.

Lines 37-38 in the revised manuscript with track changes, the nominal group has been defined in the methods paragraph of the abstract.

Point 6. Abstract - Results: is there any information on validity and reliability of the definitions. Did the steering committee vote on each definition?

This work focused on the feasibility of generating an explicit list of criteria, and the steering committee and nominal groups validated the explicit nature of the proposed definitions. The nominal group technique can generate new proposals, but consensus on the validity of definitions and their use in practice remains limited, as the nominal group technique requires small groups of participants. Baclet et al. (2022) [https://doi.org/10.1016/j.idnow.2022.02.004] set up a qualitative study to Explicit definitions of potentially inappropriate prescriptions of antibiotics in hospitalized older patients and validated this list using a delphi survey in 2024 [https://doi.org/10.3390/ antibiotics13030283].

Consequently, this study did not provide results on the validity or reliability of the definitions. The validity of these definitions will be investigated on a larger panel of participants, using a consensus method such as the Delphi method.

Point 7. Abstract - Conclusion: this is very overstated- the conclusion should be that this is a great pilot effort- and should be redone in more settings in France for validation and potential pilot data. Is a Delphi panel the next step or is a program that included more nominal groups needed?

Thank you for your suggestions regarding the conclusion of the abstract. As described in our protocol, obtaining and validating this list of definition is a 3-step process [https://doi.org/10.3390/healthcare9111539]. The Delphi method will be the next step in this study, as the delphi survey is an appropriate way to reach consensus and allows for the involvement of many participants. We modified the conclusion of the abstract in the revised version of the manuscript.

Modifications: 

Lines 49-52 in the revised manuscript with track changes, this qualitative study develop a specific list of explicit definitions of potentially inappropriate prescriptions of antidiabetic drugs in patients with type 2 diabetes mellitus by gathering the opinions of healthcare professionals caring for patients with type 2 diabetes. Although the new list provides valuable insights, it will be validated by expert consensus in a Delphi survey before being implemented in practice.

Point 8. Abbreviations- there are too many here…. Please try to decrease the number of abbreviations.

Thanks for your comment, we've reduced the number of abbreviations used.

Modifications: We have removed the 6 following abbreviations: 

ADA: American Diabetes Association, ATC: Anatomical Therapeutic Chemical, EASD: European Association for the Study of Diabetes, SFD: Société francophone du diabète, SmPC: summary of product characteristics and CDSS: clinical decision support system

We now use only 7 abbreviations instead of 13:

• AD: antidiabetic

• DDP-4: dipeptidyl-peptidase-4

• eGFR: estimated glomerular filtration rate

• GLP-1 RA: inhibitors, glucagon-like peptide 1 receptor agonist

• PIP: potentially inappropriate prescription

• SGLT2: sodium-glucose cotransporter-2

• T2DM: type 2 diabetes mellitus

Point 9. Introduction- again—too many abbreviations. 

Thanks for your comment.

Modifications: we've reduced the number of abbreviations used in the text.

Point 10. The description of implicit and explicit needs more context and references. Many studies formally compare explicit and implicit evaluations and should be cited…These studies discuss pros and cons of each method.

We thank the reviewer for this comment. In clinical pharmacology, two approaches can be used to assess inappropriateness of prescribing: the first is based on so-called 'implicit' expert judgement, and the second uses explicit criteria. An implicit judgement is based on an expert's assessment taking into account research data, clinical circumstances and patient preferences [https://doi.org/10.1007/s40266-018-0554-2]. In contrast, explicit criteria are based on predefined rules for analysing drug prescriptions and do not require the intervention of an expert. 

Implicit approach can use standardized tools, such as Medication Appropriateness Index (MAI) [https://doi.org/10.1007/s40266-013-0118-4]. This tool requires the experts to follow several steps, including (…). MAI has demonstrated relevant (Hanlon), but RCT are still missing for implicit approaches (https://doi.org/10.1007/s40266-018-0554-2). Explicit approaches, on the other hand, can use various lists or CDSS tools. The effectiveness of explicit tools such as lists of potentially inappropriate prescriptions has been well documented [https://doi.org/10.2147/DHPS.S303101. Several tools were used to assess PIM in hospitalized patients over 65most commonly Beer's criteria and the STOPP/START tool. A systematic review of the literature by Alshammari et al. reported reductions in PIM ranging from 3.5% to 87% [https://doi.org/10.2147/DHPS.S303101]. We also included references to the work of Hill-Taylor et al. [https://doi.org/10.1111/jcpt.12059], Lopez-Rodriguez et al. [https://doi.org/10.1371/journal.pone.0237186].

Explicit and implicit approaches are considered complementary. Lopez-Rodriguez [https://doi.org/10.1371/journal.pone.0237186] showed that explicit criteria are easier to implement and easier to automate. The REMEDIES list proposed a mixed approach, combining implicit and explicit recommendations. Lastly, the IGRIMUP recommended to combine these approaches [https://doi.org/10.1007/s40266-018-0554-2].

In line with the reviewer’s comment, we added some sentences to give more context and references for implicit and explicit approaches in the introduction section :

Modifications:

Lines 88-99 in the revised manuscript with track changes, we have added the parts in bold in the following text:

The MAI is the most widely accepted implicit method. However, the implicit approach has practical limitations, as itis time-consuming and requires specialized knowledge and skills to be applied.

It is also possible to reduced PIPs by applying an explicit approach, i.e. explicit criteria that combine the various items of information related to the prescription in the absence of an expert assessment using predefined criteria (e.g. a prescription of metformin in a patient with an estimated glomerular filtration rate (eGFR) below 30 mL/min/1.73 m² is a PIP). The effectiveness of this explicit approach in detecting PIPs has been well demonstrated.

These two approaches, implicit and explicit, should be combined for optimal patient management. With the increasing use of computerized physician order entry and electronic medical records, the explicit approach becomes more practical and easier to implement compared to integrating implicit criteria.

Point 11. Methods - The context for study design needs more descriptions as well--- in the introduction or methods. More text on when to use COREQ is also needed. Is there background about how many persons should be in the nominal groups and how many should be on the steering committee. Are there previous studies that have used the nominal group methods—what did they do and what did they find?

Thanks for your comment, we've clarified the context of our study design at the end of the introduction.

I

---

## [Decision Letter · Decision Letter 1]

14 Jun 2024

PONE-D-24-07859R1Development of explicit definitions of potentially inappropriate prescriptions for antidiabetic drugs in patients with type 2 diabetes: a multidisciplinary qualitative study.PLOS ONE

Dear Dr. Gerard,

Thank you for submitting your manuscript to PLOS ONE. After careful consideration, we feel that it has merit but does not fully meet PLOS ONE’s publication criteria as it currently stands. Therefore, we invite you to submit a revised version of the manuscript that addresses the points raised during the review process.

**Dear authors**I propose that both the abstract conclusion and the text conclusion explicitly state the necessity of validating the 38 explicit definitions by healthcare experts from various settings across France before proceeding to the Delphi survey. On line 287 of the manuscript conclusion it is stated "The most relevant and useful definitions for clinical practice must now be selected by expert consensus to produce a tool that can help non-diabetologists prescribe drugs". The requirement to pick "Must now" should be eliminated, as this qualitative study solely proposes explicit concepts that necessitate additional confirmation.==============================

We look forward to receiving your revised manuscript.

Kind regards,

Sairah Hafeez Kamran, PhD

Academic Editor

PLOS ONE

Reviewers' comments:

Reviewer's Responses to Questions

**Comments to the Author**

1. If the authors have adequately addressed your comments raised in a previous round of review and you feel that this manuscript is now acceptable for publication, you may indicate that here to bypass the “Comments to the Author” section, enter your conflict of interest statement in the “Confidential to Editor” section, and submit your "Accept" recommendation.

Reviewer #2: All comments have been addressed

Reviewer #3: (No Response)

2. Is the manuscript technically sound, and do the data support the conclusions?

Reviewer #2: Yes

Reviewer #3: Partly

3. Has the statistical analysis been performed appropriately and rigorously? 

Reviewer #2: Yes

Reviewer #3: No

4. Have the authors made all data underlying the findings in their manuscript fully available?

Reviewer #2: Yes

Reviewer #3: Yes

5. Is the manuscript presented in an intelligible fashion and written in standard English?

Reviewer #2: Yes

Reviewer #3: No

6. Review Comments to the Author

Reviewer #2: (No Response)

Reviewer #3: The manuscript entitled 'Development of explicit definitions of potentially inappropriate prescriptions for

antidiabetic drugs in patients with type 2 diabetes: a multidisciplinary qualitative study' provides some data about definitions of PIP in antidiabetics treatment by professionals, yet the expert opinions limited to certain region of the world may not be enough to provide a sounding definitive conclusion.

Line 51 states that 'prescription of antidiabetic drugs (ADs) can be challenging for non-specialist physicians'. This statement appear biased as prescription of antidiabetic drugs is generally challenging and needs involvement of different professionals particularly in the presence of comorbid conditions.

It would be advisable to correct grammatical errors through out the document.

For e.g

Line 60 grammar error 'expert'' expert's

Line 64 grammar error 'to reduced' to reduce

Line 153 grammar error 'was validated' were validated

7. PLOS authors have the option to publish the peer review history of their article (what does this mean?). If published, this will include your full peer review and any attached files.

Reviewer #2: No

Reviewer #3: No

---

## [Author Response · Author response to Decision Letter 1]

16 Jul 2024

Response to Journal Requirements:

We thank Academic Editor for comments and constructive suggestions. We have carefully revised the manuscript according to each comment and suggestion.

Dear authors

I propose that both the abstract conclusion and the text conclusion explicitly state the necessity of validating the 38 explicit definitions by healthcare experts from various settings across France before proceeding to the Delphi survey. 

On line 287 of the manuscript conclusion it is stated "The most relevant and useful definitions for clinical practice must now be selected by expert consensus to produce a tool that can help non-diabetologists prescribe drugs". The requirement to pick "Must now" should be eliminated, as this qualitative study solely proposes explicit concepts that necessitate additional confirmation.

We thank the editor for this comment. The validation of the 38 explicit definitions is currently ongoing through a Delphi survey among healthcare experts from various settings across France. This survey is indeed providing additional confirmation and content to the definitions (e.g. rephrasing, additional details, vote). In line with the Editor’s comment, we changed the conclusion of the abstract and in the conclusion of the article.

Modifications in line 46 of the revised manuscript with track changes: “This list of 38 explicit definitions necessitates additional confirmation by expert consensus before use in clinical practice.”

Modifications in line 328 (previous line 287) of the revised manuscript with track changes: 

• We removed the sentence: “The most relevant and useful definitions for clinical practice must now be selected by expert consensus to produce a tool that can help non-diabetologists prescribe drugs”

• We replace by the following sentence: “This list of 38 explicit definitions necessitates additional confirmation by expert consensus before use in clinical practice.”

Response to Reviewer 3 Comments

We thank Reviewer for comments and constructive suggestions. We have carefully revised the manuscript according to each comment and suggestion.

Point 1. The manuscript entitled 'Development of explicit definitions of potentially inappropriate prescriptions for antidiabetic drugs in patients with type 2 diabetes: a multidisciplinary qualitative study' provides some data about definitions of PIP in antidiabetics treatment by professionals, yet the expert opinions limited to certain region of the world may not be enough to provide a sounding definitive conclusion.

We thank the reviewer for this comment. We agree that expert opinion was limited to certain regions of the world. Such an approach has been used to develop most of the lists of potentially inappropriate prescriptions for older people. Yet, this approach did not limit the use of such lists to a single region of the world. For example, the Beers criteria were initially developed in a specific region of the USA, but have been used throughout the world. The same is true of the STOPP/START criteria (Ireland) or the FORTRA list (Germany), which are used far beyond their own countries. More importantly, the first lists of potentially inappropriate prescriptions for older people have attracted the attention of many researchers and clinicians, who have adapted the various lists to their local context. 

As a result, we believe that the first movement towards explicit definitions of potentially inappropriate prescribing of antidiabetic drugs in patients with type 2 diabetes may lead to solid advances in research in this area.

Point 2. Line 51 states that 'prescription of antidiabetic drugs (ADs) can be challenging for non-specialist physicians'. This statement appear biased as prescription of antidiabetic drugs is generally challenging and needs involvement of different professionals particularly in the presence of comorbid conditions.

We thank the reviewer for this comment. We agree, the management of patients with type 2 diabetes should ideally involve different professionals. Prescribing diabetes treatments is therefore a global challenge for all healthcare professionals, including those who are not diabetes specialists or when management is carried out in an isolated, non-pluri-disciplinary context.

In line with the reviewer’s comment, we proposed the following modifications in lines 60-62 (previous line 51) of the revised manuscript with track changes: “Managing diabetes is a challenge for all healthcare professionals, including those who are not diabetes specialists or when patients are managed in an isolated, non-multidisciplinary context.”

Point 3. It would be advisable to correct grammatical errors throughout the document.

For e.g

Line 60 grammar error 'expert'' expert's

Line 64 grammar error 'to reduced' to reduce

Line 153 grammar error 'was validated' were validated

We thank reviewer for this comment. We had the manuscript proofread by a translator (David Fraser, Biotech Communication SARL, Ploudalmézeau, France) for the initial submission. We then added some new sentences after the first review, but we did not have enough time to perform English editing with a professional translator. This has been done for this revised version of our manuscript (David Fraser, Biotech Communication SARL, Ploudalmézeau, France).

The 11 approved modifications can be found in the following sections of the revised manuscript with track changes:

• Lines 33-34: “A nominal group technique is a structured method that encourages all the participants to contribute and makes it easier to reach an agreement quickly”

• Lines 43-47: “The results of our qualitative study show that it is possible to develop a specific list of explicit definitions of potentially inappropriate prescriptions of antidiabetic drugs in patients with type 2 diabetes mellitus by gathering the opinions of healthcare professionals caring for these patients. This list of 38 explicit definitions necessitates additional confirmation by expert consensus before use in clinical practice”

• Lines 54-57: “Over the last decade, the management of patients with type 2 diabetes mellitus (T2DM) has undergone many changes; for example, a number of new antidiabetic drugs (ADs, including SGLT-2 inhibitors, GLP-1 receptor agonists, and DPP-4 inhibitors) have been approved as first-line treatments (either as monotherapies or in combination with other drugs). [1–5]. Managing diabetes is a challenge for all healthcare professionals, including those who are not diabetes specialists or when patients are managed in an isolated, non-multidisciplinary context.”

• Lines 66-69: “A professional consensus and clinical practice guidelines on the follow-up of patients with diabetes have been updated recently [6]. Patients with T2DM are mainly followed up by general practitioners [7–11]. However, various studies have shown that even after these recent updates, compliance with the guidelines is suboptimal – notably for patients followed up by general practitioners”

• Lines 75-95: “This approach is defined as implicit because it requires an expert’s assessment of the quality of care in relation to the patient’s condition and the medical literature [17, 18]. The Medication Appropriateness Index is currently the most widely used implicit method [18, 19]. However, implicit approaches have practical limitations because their application is time-consuming and requires specialized knowledge and skills.

It is also possible to reduced PIPs by applying an explicit approach, i.e. predefined explicit criteria that combine the various items of information related to the prescription in the absence of an expert assessment (e.g. a prescription of metformin in a patient with an estimated glomerular filtration rate (eGFR) below 30 mL/min/1.73 m² is a PIP) [19, 20]. The value of explicit approaches for the detection PIPs has been well documented [21–24].

Some experts recommend that implicit and explicit approaches should be combined for optimal patient management [24, 25].”

• Lines 110-113: “The nominal group technique has been widely used to gather the opinions of healthcare professionals and to generate new ideas [40–45].”

• Lines 119-121: “The present work involved a qualitative approach that had already been used to set up explicit definitions of PIP-ADs in hospitalized older patients”

• Lines 127-131: “These checklists relate to the sampling method, data collection setting, data collection method, respondent validation of findings, method of recording data, description of the derivation of themes, and inclusion of supporting quotations. The items are grouped into three domains: (i) research team and reflexivity, (ii) study design and (iii) data analysis and reporting.”

• Lines 141-145: “The inappropriateness of drug prescriptions can be evaluated in clinical pharmacology using two approaches: the first is based on a so-called “implicit” expert judgement, and the second uses explicit criteria. An implicit judgement is based on an expert’s evaluation of the quality of care with regard to the patient’s situation and guidelines on the use of drugs. In contrast, explicit criteria are based on predefined rules for the analysis of drug prescriptions, and do not require intervention by an expert.”

• Lines 152-155: “Explicit definitions cover situations considered by experts to be generally inappropriate, as defined in the literature or by expert consensus. However, when an explicit definition is applied to a given prescription, the absence of expert opinion means that the prescription’s inappropriateness cannot be confirmed. Therefore, explicit definitions correspond to PIPs.”

• Line 324-328: “Our qualitative study involved 30 healthcare professionals (10 diabetologists, 6 general practitioners, and 14 pharmacists) and generated 38 explicit definitions of PIP-ADs classified with regard to four contexts: (i) the potential need to temporarily discontinue a medication in the event of acute illness, (ii) the potential need to adjust the dosage regimen, (iii) inappropriate treatment initiation, and (iv) the need for additional monitoring of a patient with T2DM.

---

## [Decision Letter · Decision Letter 2]

9 Aug 2024

Development of explicit definitions of potentially inappropriate prescriptions for antidiabetic drugs in patients with type 2 diabetes: a multidisciplinary qualitative study.

PONE-D-24-07859R2

Dear Dr. Gerard,

We’re pleased to inform you that your manuscript has been judged scientifically suitable for publication and will be formally accepted for publication once it meets all outstanding technical requirements.

Kind regards,

Sairah Hafeez Kamran, PhD

Academic Editor

PLOS ONE

Reviewer's Responses to Questions

**Comments to the Author**

1. If the authors have adequately addressed your comments raised in a previous round of review and you feel that this manuscript is now acceptable for publication, you may indicate that here to bypass the “Comments to the Author” section, enter your conflict of interest statement in the “Confidential to Editor” section, and submit your "Accept" recommendation.

Reviewer #4: All comments have been addressed

2. Is the manuscript technically sound, and do the data support the conclusions?

Reviewer #4: Yes

3. Has the statistical analysis been performed appropriately and rigorously? 

Reviewer #4: Yes

4. Have the authors made all data underlying the findings in their manuscript fully available?

Reviewer #4: Yes

5. Is the manuscript presented in an intelligible fashion and written in standard English?

Reviewer #4: Yes

6. Review Comments to the Author

Reviewer #4: (No Response)

7. PLOS authors have the option to publish the peer review history of their article (what does this mean?). If published, this will include your full peer review and any attached files.

Reviewer #4: **Yes: **Saad Salman

---

## [Editor Report · Acceptance letter]

17 Sep 2024

PONE-D-24-07859R2 

PLOS ONE

Dear Dr. Gerard, 

I'm pleased to inform you that your manuscript has been deemed suitable for publication in PLOS ONE. Congratulations! Your manuscript is now being handed over to our production team.

Kind regards, 

on behalf of

Dr. Sairah Hafeez Kamran 

Academic Editor

PLOS ONE